# Determinants of sleep quality among pregnant women in a selected institution in the Southern province, Sri Lanka

**M. S. K. Peiris**, **Thamudi Darshi. Sundarapperuma**\*

Department of Nursing, Faculty of Allied Health Sciences, University of Ruhuna, Galle, Sri Lanka

\* Thamudids@ahs.ruh.ac.lk

## Abstract

### Background

Sleep is a vital requirement during pregnancy for the betterment of the fetus and the mother. Sleep quality could vary due to pregnancy-specific psychological and physiological changes. To introduce a tailored programme to enhance the sleep quality of mothers, it is paramount to assess the sleep quality and determinants of sleep. Therefore, this study aimed to assess the determinants of sleep quality among pregnant women in a selected institution in the Southern province of Sri Lanka.

### Methods

Hospital-based cross-sectional study was carried out with 245 antenatal women, selected using a systematic random sampling method. A pretested self-administered questionnaire was used to collect data which contains four parts. Below variables were involved and both continuous and categorical data were collected as required. 'Maternal sleep quality, socio-demographic data and gestational age, maternal depression and anxiety.' Data were analyzed using IBM SPSS version 25.0 for Windows by using descriptive statistics, Pearson's Chi-square test, and independent sample T-test (p < 0.05). Logistic regression analysis was used to find the relationship with sleep quality and other variables. P-value of less than 0.05 was considered statistically significant, at 95% CI.

### Results

The majority of women (60.8%) had good sleep quality and they didn't have either depressive symptoms (63.4%) or anxiety (64.2%). Aged between 34–41 years and third-trimester women had higher rates of poor sleep quality. Varying quality of sleep was identified among three-trimesters with subjective sleep quality, sleep latency, habitual sleep efficiency, and sleep disturbances. In comparison with the first and second trimester, pregnant women in the third trimester had higher score of global PSQI (5.22 ± 2.35), subjective sleep quality (1.23 ± 0.70), sleep latency (1.25 ± 0.86), habitual sleep efficiency (0.14 ± 0.43), and sleep disturbances (1.39 ± 0.58). There was a significant association between gestational age (P = .006), maternal age (P = .009), antenatal depression (P = .034), and anxiety (P = .013)

**Data Availability Statement:** The data set has been deposited in the OSF database (DOI 10.17605/OSF.IO/APYVK).

**Funding:** The author(s) received no specific funding for this work.

**Competing interests:** The authors have declared that no competing interests exist.

with sleep quality. However, multinomial logistic regression revealed that only gestational age affected on quality of sleep. The first trimester was a protective factor for good quality sleep (Adjusted OR = 3.156) compared to the other two trimesters.

## Conclusion

This study revealed that the majority of women had good sleep quality but quality of sleep was deprived with gestational age. It is expected that the findings of this research will be helpful for health and social care policymakers when formulating guidelines and interventions regarding improving the quality of sleep among pregnant women in Sri Lanka.

## Introduction

Sleep is one of the most important basic needs of the human being which helps to maintain metabolic activities effectively [1]. There are four elements used to measure sleep quality such as sleep latency, sleep waking, wakefulness, and sleep efficiency [2]. Pregnancy is a unique period of a woman's life, hence quality sleep is an essential requirement for a healthy pregnancy to conserve energy for the delivery process with 7–9 hour sleep [3, 4].

Poor sleep quality enhances the 20% chances of undergoing lower segment cesarean sections and prolonged labor. Further, it is a potential risk factor for mental disorders during the perinatal and postnatal period, especially depression and it could lead to postpartum psychosis [5].

The nature of the sleep during pregnancy can be varied. These changes may be unique to each trimester since during these three trimesters, specific physiological and psychological changes occur relevant to each trimester. The first trimester is considered as 1st week from the last regular menstrual period to the 12th week. In this trimester, a woman's body undergoes many hormonal changes and it triggers symptoms such as extreme tiredness, swollen breasts, mood swings, headache, and weight gain. Though most of these discomforts diminish with the progress of pregnancy, they may negatively affect sleep. The second trimester is considered as the 13th week to the 28th week. In this period, a woman can notice the enlargement of the abdomen as the fetus continues to grow. The third trimester is considered as the 29th week to the 40th week. In this period, women can experience breathing difficulties due to enlarging abdomen, heartburn, trouble sleeping, frequent movements of the fetus, and false uterine contractions with most of the second-trimester symptoms [6]. These types of factors related to each trimester can be the reasons for a frequent awakening from the sleep and poor sleep quality of a pregnant woman [7].

Physiological changes mainly build up to achieve homeostatic equilibrium for the growth of the fetus. High beta-human chorionic gonadotropin (HCG) and progesterone may affect the balancing of the duration of the REM (Rapid Eye Movement) sleep stage and non-REM (Non-Rapid Eye Movement) stage for mothers. Estrogen may influence high secretions and vascularity of the respiratory system which results in shortness of breath. High cortisol levels may activate sympathetic tone and cause to high maternal heart rate. Collectively, these changes contribute to reducing sleep duration, frequent awakening at night, snoring, and gasping [8].

A recent study has found that external factors such as low income, and poor quality of life; hence contributing to poor quality of sleep can enhance the impact of physiological changes [9]. Further, it was reported that increased age, advanced education, occupational status, gestational age, and parity-like factors also affect the sleep quality of pregnant women [10].

Therefore, depressive and anxiety symptoms are commonly associated with sleep disturbances in the pregnancy period [11]. This relationship between sleep quality and depression can cause maternal morbidity and neonatal birth complications [12].

Poor sleep quality has a high prevalence and related complications throughout the pregnancy but it is a neglected and hidden factor among healthcare providers and health policy-makers, especially in middle-income countries.

Although several studies have been conducted in Western countries to assess sleep qualities and determinants during the pregnancy period, still it is a new field for Asian countries as well as for low and middle-income countries including Sri Lanka. Further, none of the reported studies in the South Asian region had compared the differences in sleep qualities and similar results among three trimesters and neither of them had identified associated factors for the poor quality of sleep during each trimester. Therefore, the objectives of this study were to identify the prevalence of poor sleep quality, identify factors associated with poor sleep quality, compare the difference in sleep quality among three trimesters, and identify the psychological disturbances encountered by pregnant mothers due to poor sleep quality in Teaching Hospital Mahamodara (THM), Galle.

## Materials and methods

### Study setting and study population

A hospital-based descriptive-analytical study was carried out at a selected institute in Southern province of Sri Lanka. This institute was the Teaching Hospital Mahamodara, which is the only maternity teaching hospital in Southern Province, Sri Lanka. It consisted of three antenatal wards and three separate antenatal clinics attached to these wards. Approximately 3000 antenatal women registered to the clinic for a month and around 600 antenatal women were admitted to the antenatal wards.

### Eligibility criteria

Pregnant women who could understand Sinhala or English and who were willing to participate were included in the study after providing detailed information and obtaining consent. Pregnant women who were not willing to participate and who had severe mental illnesses (Psychological problems that severely impaired on person's ability to engage in functional and occupational activities such as bipolar disorders, schizophrenia etc.) [13]. were excluded from the study.

### Sample size determination and sampling procedure

The sample size was calculated by using the formula suggested by Lowanga & Lameshow in 1991 with 1.96 as the standard normal deviation for the chosen confidence level, 83% as the expected proportion of subjects with characteristics, based on the research study of public health centers in Yogyakarta, Indonesia [1] and another research study of determinants of sleep quality among pregnant women in China [14]. Accordingly, the required sample size was 217.

A further adjustment to the sample size was made considering a non-response or incomplete response rate of 10%, making the final sample size 242. There was a separate clinic register for all antenatal women. The clinic days of the week were divided into three antenatal wards and there was a separate register to admit the antenatal women to the wards. Based on these registers, women were selected by a systematic random sampling technique; individuals who belonged to the even number. To ensure the diversity of the sample, women were selected

from both antenatal clinics and wards. Fifteen to twenty-five mothers per day were selected and within 20 days sample size has been achieved.

## Data collection and management

To enhance the reliability and validity of the study, several standard steps were taken throughout the study. Questionnaires were prepared in both languages and women were allowed to select the most convenient language to fill out the questionnaire. Further, all the questionnaires were prepared in simple language without medical jargon. All were pretested with similar subjects in a different setting and all the ambiguous statements and interpretation errors were corrected. Most women had been educated up to the Ordinary Level or below, taking to the account that Sri Lankan literacy level is 92.25% [15].

The appropriate sample was taken to the study with considering the 10% non-respondent rate to enhance the validity of the results.

The self-administered questionnaire which consisted of four parts was used to collect data.

Part A- Self developed part to assess socio-demographic data and pregnancy-related information.

Part B- Sinhala version of the validated Pittsburg Sleep Quality Index (PSQI) tool was used to assess the quality of sleep.

Part C- Sinhala version of the Edinburgh postnatal depression scale which was a widely validated tool to assess depression throughout the pregnancy and postpartum was used to identify the depression status.

Part D- Sinhala version of the General Anxiety Disorders 7 (GAD 7) scale was used to assess the anxiety status of the mothers. However, GAD 7 was not a cross-cultural validated tool hence content and face validity were assessed before applying it.

This questionnaire was a pretest with ten pregnant mothers in another setting in the Galle district to prevent participants' contamination.

To reduce the socially desirable bias and subjective bias participants were asked to complete this questionnaire by themselves. However, trained investigators were there to give any assistance.

The average duration of the questionnaire was 15 minutes.

All the respondents participated in the study voluntarily. Hence, the response rate of the study was 100%. The incomplete questionnaires were removed from the study and a new response was collected until the sample size was achieved.

## Operational definitions

Sleep quality, depression, and anxiety were identified as operational definitions of this study.

Sleep quality—Sleep quality was assessed by the Pittsburg Sleep Quality Index (PSQI). It was a cross-cultural validated tool for Sri Lankans with good internal consistency (Cronbach's alpha = 0.85). It consisted of 19 self-evaluated questions under seven components i.e. subjective sleep quality, sleep latency, sleep duration, habitual sleep efficiency, sleep disturbances, use of sleep medications, and daytime functions. These questions are scored to get a total mark range from 0–21 and a higher scale shows severe difficulty in all areas [16].

Depression status—Depression status was assessed by the Edinburgh Postnatal Depression Scale (EPDS). It is also a validated questionnaire for Sri Lankan antenatal mothers (Cronbach's alpha = 0.95). It consists of 10 questions. Each answer is given a score from 0 to 3. The maximum score is 30 and higher indicates severe depressive status [17].

Anxiety level—Anxiety level was screened by the General Anxiety Disorder 7 scale (GAD-7). It consists of 7 items and each item was rated by 4-point Likert scale. Though it was a widely

validated tool in several countries, still it was not a validated tool in Sri Lanka. Hence content and face validity of this questionnaire were assessed before applying it [18].

## Data analysis

Data was analyzed using IBM SPSS (Statistical Package for Social Sciences) 25 version software. Assumptions and requirements, such as linearity and normality were examined in the data set, as these were prerequisites for the conduction of parametric tests [19]. Descriptive statistics such as mean (SD), median (IQR), and frequencies (%) were used to describe the level of quality sleep and other variables for the study of the participants. Normally distributed data was further analyzed for inferences, using an independent samples t-test and ANOVA. Chi-square analysis was also used to find the association between categorical variables. The independent variables that were observed as significant in the chi-square, paired sample T-test, or ANOVA test were tested again for significance with univariate logistic regression. Multinomial logistic regression was used to determine the exact factors associated with good sleep quality while removing the weak relationships and maintaining only the strong relationships. The risk and protective factors were determined by generating the Adjusted Odds Ratio (OR) with a 95% Confidence Interval (CI). P value less than 0.05 ($p<0.05$) was considered as statistically significant. PSQI, EPDS and GAD-7 scales were analyzed following the guidelines provided by the respective tool developers [18, 20].

## Ethics approval and consent to participate

Ethical clearance was obtained from the Ethical Review Committee, Faculty of Allied Health Sciences, University of Ruhuna (No: 59.11.2021). Informed written consent was taken from all participants after providing detailed information about the purpose of the research study, advantages, results, confidentiality, and methods of the research. Confidentiality of all information was kept properly. The data were collected from 12th July to 21st July 2022.

## Results

The majority of subjects were married (98.8%), unemployed (73.1%), and educated up to Ordinary Level—O/L (43.3%) with monthly income between 30,000–40,000 rupees. Among the sample, most of the pregnant women were in the third trimester (48.2%) and had children between 1–3 children (57.1%). Also, 72.7% of participants had normal weight (18.5 to 24.9 Kg) according to the Body Mass Index (BMI) (Table 1).

### Association between sleep quality and basic factors

The mean Global PSQI showed a significant association between maternal age and gestational age (Table 2).

### Sleep quality among trimesters

The results showed that there were 39.2% had poor sleep quality while 60.8% had good sleep quality (Table 3) with mean global PSQI score 4.8531 (SD = 2.3053)

### The sleep pattern of pregnant women

Frequency distribution of sleep pattern according to the PSQI shows that 65.3% of pregnant women had taken less than 15 minutes to fall asleep each night during the past month while 39.6% of mothers had not any trouble during the last month to sleep within 30 minutes.

**Table 1. Socio-demographic and other relevant characteristics of the participants.**

| Variables | Subcategories | Frequencies (%) |
|---|---|---|
| Age in Years | 18–25 | 62 (25.3) |
| | 26–33 | 126 (51.4) |
| | 34–42 | 57 (23.2) |
| Occupational Status | Employed | 66 (26.9) |
| | Unemployed | 179 (73.1) |
| Educational Status | Degree/higher degree/Diploma/other vocational programme | 46 (18.5) |
| | Up to Advanced Level (A/L) | 77(31.4) |
| | Up to Ordinary Level (O/L) | 122 (49.8) |
| BMI (kg/m$^2$) | Underweight (<18.5) | 33 (13.5) |
| | Normal weight (18.5–24.9) | 178 (72.7) |
| | Overweight (25–29.9) or obese (≥30) | 34 (13.9) |
| Gestational Age | Week 1–12 | 55 (22.4) |
| | Wee 13–26 | 72 (29.4) |
| | Week 27–40 | 118 (48.2) |
| Number of children | 0 | 99 (40.4) |
| | 1–3 | 140 (57.1) |
| | ≥4 | 6 (2.4) |
| Monthly Income | < LKR 30000.00 | 69 (28.2%) |
| | LKR 30001–40000.00 | 99 (40.4%) |
| | >40000.00 | 77 (31.4%) |

**Table 2. Association between sleep quality and basic factors (n = 245).**

| Variables | Subcategories | Frequencies (%) | F-test | p-value |
|---|---|---|---|---|
| Age in Years (**) | 18–25 | 62 (25.3) | 3.96 | **0.01** |
| | 26–33 | 126 (51.4) | | |
| | 34–42 | 57 (23.2) | | |
| Occupational Status (*) | Employed | 66 (26.9) | 2.68 | 0.10 |
| | Unemployed | 179 (73.1) | | |
| Educational Status (**) | Degree/higher degree/Diploma/other vocational programme | 46 (18.8) | 0.08 | 0.51 |
| | | 16 (6.5) | | |
| | | 12 (4.9) | | |
| | Up to Advanced Level (A/L) | 77(31.4) | | |
| | Up to Ordinary Level (O/L) | 122 (49.8) | | |
| BMI (kg/m$^2$) (**) | Underweight (<18.5) | 33 (13.5) | 1.63 | 0.18 |
| | Normal weight (18.5–24.9) | 178 (72.7) | | |
| | Overweight (25–29.9) or obese (≥30) | 34 (13.9) | | |
| Gestational Age (**) | Week 1–12 | 55 (22.4) | 5.47 | **0.01** |
| | Wee 13–26 | 72 (29.4) | | |
| | Week 27–40 | 118 (48.2) | | |
| Number of children (**) | 0 | 99 (40.4) | 0.03 | 0.96 |
| | 1–3 | 140 (57.1) | | |
| | ≥4 | 6 (2.4) | | |
| Income | < LKR 30000.00 | 69 (28.2%) | 2.697 | |
| | LKR 30001–40000.00 | 99 (40.4%) | | |
| | >40000.00 | 77 (31.4%) | | |

(*) used Independent sample T-test

(**) used one-way ANOVA test

**Table 3. Distribution of sleep quality among trimesters (n = 245).**

| Gestational Age | Sleep Quality | | P value |
|---|---|---|---|
| | Good Sleep Quality 149 (60.8%) | Poor Sleep Quality 96 (39.2%) | |
| Week 1–12 | 43 (28.9%) | 12 (12.5%) | 0.01 |
| Week 13–26 | 44 (29.5%) | 28 (29.2%) | |
| Week 27–40 | 62 (41.6%) | 56 (58.3%) | |

Chi-square P ≤0.05

Also, the frequency distribution of sleep duration according to the PSQI indicated 77.1% of pregnant women had more than 7 hours of sleep during the last month while 86.9% of pregnant women had more than 85% of habitual sleep efficiency during the last month. (Table 4).

## PSQI score and the differences among three trimesters

The results showed that pregnant women in the third trimester had higher scores of global PSQI, subjective sleep quality, sleep latency, habitual sleep efficiency, and sleep disturbances than first and second trimesters. When compared to the third trimester, pregnant women in the second trimester had higher scores of sleep duration, habitual sleep efficiency, and daytime dysfunction. There were no statistically significant differences across trimesters in the following PSQI component scores: sleep duration, use of sleeping medication, and daytime dysfunction (Table 5).

## Psychological disturbances and sleep quality

The results showed that there was a significant association between antenatal depressions with sleep quality (p < 0.05). Poor sleep quality was experienced by more than half of pregnant

**Table 4. Frequency distribution of sleep patterns according to the PSQI.**

| Variable | Subcategories | Frequencies (%) |
|---|---|---|
| **Time needed to fall asleep (in minutes)** | <15 minutes (0) | 160 (65.3) |
| | 16–30 minutes (1) | 66 (26.9) |
| | 31–60 minutes (2) | 15 (6.1) |
| | >60 minutes (3) | 04 (1.6) |
| **Difficulties in falling asleep within 30 min** | Not during the past month | 97 (39.6) |
| | Less than once a week | 54 (22.0) |
| | Once or twice a week | 27 (11.0) |
| | Three or more times a week | 67 (27.3) |
| **Night sleep duration** | >7 hours (0) | 189 (77.1) |
| | 6–7 hours (1) | 37 (15.1) |
| | 5–6 hours (2) | 18 (7.3) |
| | <5 hours (3) | 01 (0.4) |
| **Habitual sleep efficiency** $= \frac{Total\ of\ hours\ asleep}{Total\ of\ hours\ in\ bed} \times 100$ | >85% (0) | 213 (86.9) |
| | 75%– 84% (1) | 27 (11.0) |
| | 65%– 74% (2) | 04 (1.6) |
| | <65% (3) | 01 (0.4) |
| **Use of hypnotics** | Not during the past month (0) | 189 (77.1) |
| | Less than once a week (1) | 37 (15.1) |
| | Once or twice a week (2) | 18 (7.3) |
| | Three or more times a week (3) | 01 (0.4) |

**Table 5. PSQI score and the differences among three trimesters.**

| Components | Gestational age | | | | F-test | P-value |
|---|---|---|---|---|---|---|
| | Mean score (n = 245) | First trimester (n = 55) | Second trimester (n = 72) | Third trimester (n = 118) | | |
| **Global PSQI score** | 4.85 ± 2.30 | 4.00 ± 1.89 | 4.90 ± 2.36 | 5.22 ± 2.35 | 5.47 | 0.005 |
| **Component score** | | | | | | |
| Subjective sleep quality | 1.11 ± 0.66 | 0.91 ± 0.55 | 1.08 ± 0.64 | 1.23 ± 0.70 | 4.53 | 0.012 |
| Sleep latency | 1.06 ± 0.89 | 0.78 ± 0.81 | 0.98 ± 0.96 | 1.25 ± 0.86 | 5.88 | 0.003 |
| Sleep duration | 0.31 ± 0.62 | 0.29 ± 0.68 | 0.38 ± 0.64 | 0.27 ± 0.58 | 0.83 | 0.436 |
| Habitual sleep efficiency | 0.15 ± 0.43 | 0.05 ± 0.29 | 0.26 ± 0.50 | 0.14 ± 0.43 | 3.93 | 0.021 |
| Sleep disturbances | 1.28 ± 0.54 | 1.09 ± 0.48 | 1.25 ± 0.49 | 1.39 ± 0.58 | 5.96 | 0.003 |
| Use of sleeping medication | 0.11 ± 0.53 | 0.09 ± 0.48 | 0.09 ± 0.41 | 0.14 ± 0.61 | 0.18 | 0.832 |
| Daytime dysfunction | 0.80 ± 0.81 | 0.78 ± 0.73 | 0.83 ± 0.59 | 0.80 ± 0.82 | 0.06 | 0.939 |

women who had possible depression of varying severity (56.3%), pregnant women who did not experience depression also had much worse sleep quality (36.6%). The sleep quality of pregnant women who had severe levels of anxiety had poor sleep quality (100%), and pregnant who did not have anxiety also had poor sleep quality (35.8%). Also, this result shows that there was a significant association between anxiety with sleep quality (Table 6).

## Chi-square P≤0.05

**Factors which affected the quality of sleep.** Multinomial logistic regression revealed that only gestational age affected sleep quality (p<0.01) (Table 7).

## Discussion

The descriptive analytical study showed that pregnant women experienced specific sleep disturbances, leading to poor sleep quality during pregnancy. Surprisingly, despite the trimester, two-thirds of women in our study had declared that they have good quality sleep when assessed Pittsburg Sleep Quality Index (PSQI). The findings are slightly similar to the study with 1298 mothers in Peru [21]. Only 17% of women have poor sleep quality between 24–28 weeks. However, some studies [12, 22] have reported contraindicated findings to our study. These publications showed that considerably higher rates of sleep disturbances in pregnant women. Considering the trimesters, in our study, mothers in the first trimester showed significantly higher quality of sleep than the other two trimesters. In, logistic regression, after controlling all the cofounders, it was confirmed that mothers in the first trimester have around three times higher good quality sleep than the other two semesters. Parallel to the findings of our study, the Ethiopian study with 415 mothers has found that 69.2% mothers have good quality sleep within the first trimester [19].

**Table 6. Association of psychological disturbances and sleep quality.**

| No. | Variables | Subcategories | Sleep Quality | | X² value | p-value |
|---|---|---|---|---|---|---|
| | | | Good sleep quality n (%) | Poor sleep quality n (%) | | |
| 1. | Depression | No depression status | 135 (63.4) | 78 (36.6) | 4.499 | 0.034 |
| | | Possible depression of varying severity | 14 (43.8) | 18 (56.3) | | |
| 2. | Anxiety | No anxiety | 136 (64.2) | 76 (35.8) | 10.799 | 0.013 |
| | | Mild or high-level anxiety | 13 (20.6.0) | 20 (79.4) | | |

**Table 7. Factors impacted on sleep quality (n = 245).**

| Dependent variable | Independent variables | Standard Error | Adjusted OR (95% CI) | *p-value |
|---|---|---|---|---|
| Sleep Quality | Anxiety (no anxiety) | 0.481 | 1.85 (0.72–4.75) | 0.20 |
| | Depression (low score) | 0.042 | 0.93 (0.85–1.007) | 0.072 |
| | Gestational age (1st trimester) | 0.39 | 3.16 (1.47–6.78) | **0.003** |
| | Mother's age (18–25 years) | 0.39 | 1.52 (0.71–3.26) | 0.283 |

p-values derived from Multinomial Logistic Regression

Odds Ratio—OR, Confidence Interval–CI

Quality of sleep also varies within three trimesters of pregnancy due to fluctuation of hormonal levels throughout the body. Mainly, the Melatonin hormone helps improve sleep levels by regulating the circadian rhythms of several biological functions, including the sleep-wake cycle [2]. According to the present study, women in the third trimester experienced higher scores of global PSQI, subjective sleep quality, sleep latency, habitual sleep efficiency, sleep disturbances, and significantly longer sleep latency than women in the other two trimesters. Further, good sleep quality gradually decreased from the first trimester to the third trimester. The findings are comparable with the Taiwan study which was conducted with 400 antenatal mothers found that sleep latency, subjective sleep quality, and habitual sleep efficiency were significantly different by trimester, and women in their third trimester had significantly longer sleep latency than women in their first trimester. Hence, women in the third trimester also had significantly lower habitual sleep efficiency and subjective sleep quality than other trimesters [23].

In addition to the gestational age, several factors can be affected by poor sleep quality. A more recent study in China has found that increased maternal age is one of the important determinants of sleep quality [14]. Another Ethiopian study has found, multiparity as another determinant addition to the above factors [10]. Moreover, an association between age and sleep quality was observed in the present study. However, after running logistic regression any significance didn't find between these two variables. Further, socioeconomic factors didn't show any significance in sleep quality in this study.

Analogs to our study, the study with African Americans indicated that multiparous women experienced than nulliparous women longer sleep latency and more frequent sleep disturbances (e.g., nighttime and early morning awakenings) when it goes from the first trimester to the third trimester [24]. In the present study, the mean sleep latency score was 1.06 ± 0.89 respectively. Almost similar results were found in an Iranian study among 605 pregnant women the mean sleep latency score and a mean score of sleep disturbances gradually increased with gestational age [25]. These findings may be attributed to similar physiological changes during pregnancy. It is known that physical and mechanical factors such as frequent urination at night, a very active baby, heartburn sensation, and increased level of estrogen and oxytocin might be reasons for increased sleep disturbance during pregnancy [9].

Anxiety and depression are common psychological disturbances during pregnancy and postpartum due to the fluctuation of maternal hormones [14]. Generalized Anxiety Disorders–7 questionnaires and the Edinburgh Postnatal Depression Scale were used to assess the anxiety and level of depression among mothers respectively. The results of this present study are in line with several studies [26] around the world. A study conducted among 630 Spanish pregnant women has revealed that maternal anxiety, maternal stress, and sleep quality disturbances are prevalent throughout their pregnancy and further compromised over gestation [27]. Hence, poor sleep quality is experienced by more than half of pregnant women who have

possible depression of varying severity [28]. These findings may be attributed to the hidden fear of the impending birth of the child in women who experience pregnancy for the first time. This might be enhanced by a cultural background in our country, some pregnancy-related myths such as the whole baby care process belonging to the mother, knowledge and experience level of senior members of a family about pregnancy [29].

Sleep quality can be affected by several sociodemographic and other pregnancy-related factors. In the present study, a statistically significant association was observed between gestational age and poor sleep quality among pregnant mothers, showing that sleep quality decreases as pregnancy continues. This is in parallel with the findings of studies conducted in China [14], and Ethiopia [10]. This may be attributed to hormonal changes, increased maternal anxiety, and fetal movement in the final trimester of the pregnancy.

## Limitations and strengths of the study

There are several limitations that we have to highlight in our study. This study was conducted in a single setting in a Galle district in Sri Lanka with one ethnic group hence some generalizability issues could be found.

Sleep quality was assessed by the Pittsburgh Sleeping Quality Index, which is the validated tool for Sri Lanka and it helps to assess sleep quality by using seven components of the sleep process over one month, but it could be difficult to recall all the details related to their last months' sleep. Hence, recall bias was identified as a limitation of this study.

We didn't assess the living conditions of these mothers. However, living conditions can vary with sleeping problems. Further, we had to use separate tools to assess the living conditions but it was practically difficult to spare more time since we had a very limited period.

However, the respondent rate of this study is 100% of this study and all parts of the questionnaire were validated or pretested. Therefore, the validity and reliability of the study were at an optimal level which could be considered as one of the strengths of our study. Further, participants filled out the questionnaire by themselves. Hence, Subjective bias and socially desirability may probably be minimized.

This study was conducted in the selected institution in southern Sri Lanka. This institute, Teaching Hospital Mahamodara is the largest and only maternal institution in southern part of Sri Lanka. As this sample is a diverse sample that enriches different socioeconomic domains, the findings of this study are broadly applicable to all parts of the country since this sample represents most of the characteristics of women living in other parts of the country.

This study may probably be the first published study not only in Sri Lanka but also in South Asia which leads future guides for healthcare workers to take actions to enhance the quality of the pregnant period and to carry out future studies with larger scales.

## Conclusions

This study showed that the prevalence of poor sleep quality among pregnant women was considerably low. Advanced maternal age, gestational age, prenatal depression and anxiety were predominant determinants of sleep quality during pregnancy period. Good sleep quality gradually decreased from the first trimester to the third trimester. Women in the third trimester experienced higher habitual sleep efficiency, sleep disturbances, and significantly longer sleep latency than women in the other two trimesters. Among the major components of quality sleeping according to the PSQI, subjective sleep quality, sleep latency, habitual sleep efficiency, and sleep disturbances showed that significant difference around three trimesters. It is expected that the findings of this research will be helpful for health and social care policymakers when formulating guidelines and interventions regarding improving the quality of

sleep among pregnant women in Sri Lanka. It would be better to use a screening tool to assess the sleep quality of mothers in clinical practices to take necessary measurements to enhance the sleep quality of pregnant mothers as suggested by a recent study [30] to robust the quality of life during pregnancy.

As suggested by Smyka et al., we also suggest to discuss sleep hygiene, risk factors for poor sleep, and the importance of sleep during pregnancy with mothers and family members [26].

A large-scale study could be suggested to conduct with different ethnic groups and cultures to enhance the validity of the study.

Further, it is suggested to tailor these findings to relevant government agencies and public health policies to plan awareness programme to enhance maternal health at the primary care level and to enhance sleep quality throughout the pregnancy.

## Supporting information

**S1 Appendix. English version for demographic details questionnaire.**
(PDF)

**S2 Appendix. English version of PSQI and EPDS.**
(PDF)

**S3 Appendix. English version of GAD-7 scale.**
(PDF)

**S4 Appendix. English version information sheet.**
(PDF)

**S5 Appendix. English version consent form.**
(PDF)

## Acknowledgments

The authors would like to acknowledge the mothers for their participation and Ms. L. H. W. Dilprabha, lecturer in English, Faculty of Allied Health Sciences, University of Ruhuna, Sri Lanka for correcting the language errors.

## Author Contributions

**Formal analysis:** Thamudi Darshi. Sundarapperuma.

**Methodology:** M. S. K. Peiris, Thamudi Darshi. Sundarapperuma.

**Supervision:** Thamudi Darshi. Sundarapperuma.

**Writing – original draft:** M. S. K. Peiris.

**Writing – review & editing:** Thamudi Darshi. Sundarapperuma.

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
