## [Decision Letter · Decision Letter 0]

18 Feb 2024

PONE-D-24-00504Determinants of sleep quality among pregnant women in teaching hospital mahamodara, Galle, Sri LankaPLOS ONE

Dear Dr. Sundarapperuma,

Thank you for submitting your manuscript to PLOS ONE. After careful consideration, we feel that it has merit but does not fully meet PLOS ONE’s publication criteria as it currently stands. Therefore, we invite you to submit a revised version of the manuscript that addresses the points raised during the review process.

We look forward to receiving your revised manuscript.

Kind regards,

Trhas Tadesse Berhe, PhD

Academic Editor

PLOS ONE

2. In this instance it seems there may be acceptable restrictions in place that prevent the public sharing of your minimal data. However, in line with our goal of ensuring long-term data availability to all interested researchers, PLOS’ Data Policy states that authors cannot be the sole named individuals responsible for ensuring data access (http://journals.plos.org/plosone/s/data-availability#loc-acceptable-data-sharing-methods).

Additional Editor Comments:

**Abstract Formatting:**
Ensure that your abstract strictly adheres to our Instructions for Authors' formatting guidelines.

**Supplementary File:**
Kindly include the supplementary(like questioner / consent/assent  file as requested.

**Introduction:**
Provide a clearer explanation of the research gap and emphasize the significance of the study in the introduction.

**Source of Tool/Questionnaire:**
Specify the source of the tool/questionnaire used in your study.

**Reliability and Validity:**
Outline the steps taken to assure the reliability and validity of your data.

**Operational Definition and Measurement:**
Please provide clarification on how you measure the operational definitions. Include specific details such as the number and type of questions used to assess factors like sleep quality and anxiety."
**Categorization of Continuous Variables:**
Ensure that all continuous variables in the results section are categorized in a standard way or adapted from similar previous literature. For instance, the response to the question ‘How many hours of actual sleep did you get at night?’ was counted twice in the second and third columns. Please revisit this issue.

**Limitations and Strengths:**
Include a section on limitations and strengths to provide a comprehensive overview of your study.

**Tables and Figures:**
Adhere to the guidelines regarding tables and figures.
**Abstract Formatting:**
Ensure that your abstract strictly adheres to our Instructions for Authors' formatting guidelines.

**Supplementary File:**
Kindly include the supplementary(like questioner / consent/assent  file as requested.

**Introduction:**
Provide a clearer explanation of the research gap and emphasize the significance of the study in the introduction.

**Source of Tool/Questionnaire:**
Specify the source of the tool/questionnaire used in your study.

**Reliability and Validity:**
Outline the steps taken to assure the reliability and validity of your data.

**Operational Definition and Measurement:**
Please provide clarification on how you measure the operational definitions. Include specific details such as the number and type of questions used to assess factors like sleep quality and anxiety."
**Categorization of Continuous Variables:**
Ensure that all continuous variables in the results section are categorized in a standard way or adapted from similar previous literature. For instance, the response to the question ‘How many hours of actual sleep did you get at night?’ was counted twice in the second and third columns. Please revisit this issue.

**Limitations and Strengths:**
Include a section on limitations and strengths to provide a comprehensive overview of your study.

**Tables and Figures:**
Adhere to the guidelines regarding tables and figures.

Reviewers' comments:

Reviewer's Responses to Questions

**Comments to the Author**

1. Is the manuscript technically sound, and do the data support the conclusions?

Reviewer #1: Yes

Reviewer #2: Partly

Reviewer #3: Partly

Reviewer #4: Yes

2. Has the statistical analysis been performed appropriately and rigorously? 

Reviewer #1: Yes

Reviewer #2: No

Reviewer #3: No

Reviewer #4: Yes

3. Have the authors made all data underlying the findings in their manuscript fully available?

Reviewer #1: Yes

Reviewer #2: Yes

Reviewer #3: Yes

Reviewer #4: Yes

4. Is the manuscript presented in an intelligible fashion and written in standard English?

Reviewer #1: No

Reviewer #2: Yes

Reviewer #3: Yes

Reviewer #4: Yes

5. Review Comments to the Author

Reviewer #1: - Minor grammatical mistakes

- Data collection and management: tools validity etc are missed

- Line 130: short form used with ? meaning

- There are just 15 references, seems lack of sufficient evidences to support the research

- Some references are missing

- Line 209-210: sentence repeated, cultural difference should be further elaborated

- Line 238-241, lack of reference

- Result of similar studies in Asia and Africa are used, studies in Europe and America are also suggested to be used and compare

- Line 272-274, please further elaborate the suggestion with example and reference

Reviewer #2: Summary of the research paper:

1. The study investigated determinants of sleep quality among pregnant women in Sri Lanka, highlighting the scarcity of research in this area, particularly within the Sri Lankan context. The manuscript is well-organized, highlighting study's significance to Sri Lanka and maternal health context. Study objectives are well outlined. Although some study limitations are recognized, further improvements and clarifications are necessary to enhance the manuscript's overall contribution to public health literature.

Major Comments

2. Introduction: Authors to consider expanding on the introduction to include more recent studies from around the world to provide a comprehensive overview of evidence on sleep quality among pregnant women and its impact on pregnancy outcomes.

3. Methodology:

4. The methodology appears robust, employing a descriptive cross-sectional design with a reasonable sample size calculated based on acceptable methodology.

5. However, clarification on data collected using PSQI, EPDS, GAD-7 would be helpful to the reader. For example, it is not clear if social factors influencing sleep quality such as family support, and partner relationship characteristics, and drug and alcohol use were collected /assessed.

6. Questionnaire – a local language and English was used, yet most respondents only had O/L or below, can the authors clarify why trained data collectors were not considered to administer questionnaires?

7. Moreover, how depression and anxiety were assessed need clarification?

8. Although study limitations have been acknowledged, clarifying potential bias/limitation in the sampling process would enhance study credibility. For example, the manuscript did not outline if the influence of existing conditions likely to influence sleep quality was considered, or bias of any sort.

9. Authors to clarify study response rate, and further outline how incomplete questionnaires were dealt with.

10. Provide background information on study the setting, such as population characteristics etc., how many pregnant women are treated per year.

11. Did the authors target patients enrolled at the center during a certain period, considering data was collected in a single month, please clarify how the list was drawn up.

12. The results section mentions monthly income but population background regarding social factors such as living conditions, average household income, and employment trends in the area are not provided. Difficult to understand how the results would be interpreted in the absence of the aforementioned.

13. Authors to consider defining key concepts/terms such as A/L and O/L throughout the manuscript to improve clarity and other undefined terms. Majority of participants appear to have an education level of up to O/L but it is not clear what this means, and could authors consider merging categories with low participants, for example, categories up to A/L can be merged

Results:

14. The results indicated that 39.2% of participants experienced poor sleep quality, while 60.8% enjoyed good sleep quality, with a mean global PSQI score of 4.8531 (SD = 2.3053). Most participants reported good sleep quality during the last month of their pregnancy, according to the PSQI results. Authors to consider referencing the specific tables where these data are drawn from for clarity (Line 140 to 141).

15. To improve clarity and demonstrate consideration to scientific rigor., when claiming a significant difference, Authors to explicitly outline the P - Value (example Line 174), In addition, Authors to outline % with number of respondents where reference Is made to majority or frequency of key factors.

16. The presentation of results is consistent, and the statistical analysis performed is appropriate. However, the study could be enhanced by additional analyses. Authors to consider stratifying analysis by trimester or gestational age and conduct logistic regression to adjust for confounders, reporting odds ratios (OR) and confidence intervals (CI), which will give scientific rigor to the paper.

17. Given that gestational age showed significant variance, stratifying the analysis by gestational age could offer a deeper understanding of its impact on sleep quality.

18. Furthermore, the significance of maternal age (p-value = 0.001) suggests the need for logistic regression analysis to determine the odds ratio for poor sleep quality across different ages and trimesters. Highlighting these odds ratios would provide clearer insights into the risk factors associated with poor sleep quality during pregnancy.

19. Additionally, exploring how education and social factors, including the number of children, affect sleep quality across trimesters could enrich the study's findings.

Minor Issues

20. In the abstract, it is mentioned, "Gestational age and increasing maternal age were significantly associated with sleep quality" (Line 31). Also, see Lines 37-38. Indicating the p-value where significant differences are noted would be beneficial.

21. Line 30 states, "60.8 of participants had good sleep quality and did not experience depression and anxiety." Also, consider Lines 44-45. It is not clear what 60.8 refers to.

22. Authors to consider tailoring recommendations to relevant government agencies, stakeholders, and public health policies based on the study findings for greater impact.

23. Citation Consistency: It is noted in various sections that studies have been reported, but often only one or two studies are cited (Line 61).

24. Tables and Terminology: Tables and their titles could be improved, and terms like F and P should be defined, particularly for Table 3.

25. Ethics Discussion Repetition: Ethics-related issues are mentioned in more than one section of the manuscript, specifically on Lines 102-103.

26. Definition of Severe Psychiatric Illnesses: The authors should clarify how severe psychiatric illnesses were defined, as mentioned in Lines 79-80.

Recommendation/s:

This study has the potential to significantly contribute to maternal health and public health. However, to achieve this, major revisions are required. These should focus on expanding the methodology by clarifying details as outlined above, addressing study weaknesses and biases, enhancing the analysis, and broadening the background section to include pregnancy outcomes linked to sleep quality. Additionally, grammatical errors need correction.

Reviewer #3: This is an important area of research aimed at improving the health of pregnant women. However, there are a few comments for your attention.

The authors contribution, did the first author only contributed by collecting data? Was the first author involved in writing the manuscript?

On the abstract, the first sentence is the same as the one in a paper by Anbesaw found on: https://bmcpsychiatry.biomedcentral.com/articles/10.1186/s12888-021-03483-w. May you please rephrase it. On the same abstract, line 30, you have a value of 60.8, please indicate the units. And on data analysis, what variables were involved and what type of data was collected.

On methods, was the study only descriptive or also analytical? On sample size, line 92, is it not fine to mention the name of clinic? What is a self-developed questionnaire? Line 103, on ethical consideration should be deleted because there is a full sub section on ethical considerations. It is indicated that simple random sampling was used but you also indicate that the even numbers were selected. This is not simple random sampling. Please indicate how variables were measured. Were there scales? What type of data were used?

Line 105, the operational definitions should be included in the introduction and not stand alone. On data analysis, the SPSS should be IBM. Did you test for normality to confirm if data was normally distributed? Was the data suitable for ANOVA?

On results, the words majority, most etc are subjective. Its better to start with demographics then prevalence of poor sleep quality etc. Results on age show that age was measured as categorical data so where are the means coming from? Similar with other categorical variables. Please explained how these were measured. Table 2 has variables which look like questions taken directly from the questionnaire. Please check. You have made claims on presence of statistical significant relationship without evidence. There is a mix up on presentation of results, you present results for Table one under a different health i.e. under Table 2 or 3. Any explaination on the Global PSQI score?

On discussion, the first paragraph need to be deleted. It is important to start by answering the main research question, explain the answer and then back it up with other literature. May you please learn from other well written papers on how to write a good discussion section.

Reviewer #4: I appreciate the opportunity to review this interesting report on Determinants of sleep quality among pregnant women in teaching hospital mahamodara, Galle, Sri Lanka. I enjoyed reading the manuscript. I think this paper is excellent and is an important addition to the literature.

The paper presents a contribution to the body of knowledge especially in the field of maternal and child health especially

promoting mental health and psychological support for pregnant women by ensuring quality of sleep during pregnancy. The structure of the paper is coherent and flows logically from the introduction to the conclusion. The research undertaken is contextualized clearly. The researcher has studied and used appropriate number of bibliography sources used and quoted in the paper. It is the evidence of the deep theoretical knowledge and very good orientation in the problem discussed in the paper. The word processing of the paper is adequate. The using of different fonts and structure of the text is proper and helps the reader to better orientation in the text. The paper is in a form suitable to the discipline. The format and literary presentation of the paper are satisfactory. Writing of the document is of professional standard.

6. PLOS authors have the option to publish the peer review history of their article (what does this mean?). If published, this will include your full peer review and any attached files.

Reviewer #1: No

Reviewer #2: No

Reviewer #3: **Yes: **Save Kumwenda

Reviewer #4: No

---

## [Author Response · Author response to Decision Letter 0]

26 Apr 2024

Dear Sir/Madam,

All the comments have been addressed.

---

## [Decision Letter · Decision Letter 1]

29 May 2024

Determinants of sleep quality among pregnant women in teaching hospital mahamodara, Galle, Sri Lanka  

Changed title Determinants of sleep quality among pregnant women in a selected institution in the Southern province, Sri Lanka

PONE-D-24-00504R1

Dear Dr. Sundarapperuma,

We’re pleased to inform you that your manuscript has been judged scientifically suitable for publication and will be formally accepted for publication once it meets all outstanding technical requirements.

Kind regards,

Trhas Tadesse Berhe, PhD

Academic Editor

PLOS ONE

Additional Editor Comments (optional):

Reviewers' comments:

Reviewer's Responses to Questions

**Comments to the Author**

1. If the authors have adequately addressed your comments raised in a previous round of review and you feel that this manuscript is now acceptable for publication, you may indicate that here to bypass the “Comments to the Author” section, enter your conflict of interest statement in the “Confidential to Editor” section, and submit your "Accept" recommendation.

Reviewer #1: All comments have been addressed

Reviewer #2: All comments have been addressed

2. Is the manuscript technically sound, and do the data support the conclusions?

Reviewer #1: Yes

Reviewer #2: Yes

3. Has the statistical analysis been performed appropriately and rigorously? 

Reviewer #1: Yes

Reviewer #2: Yes

4. Have the authors made all data underlying the findings in their manuscript fully available?

Reviewer #1: Yes

Reviewer #2: Yes

5. Is the manuscript presented in an intelligible fashion and written in standard English?

Reviewer #1: Yes

Reviewer #2: Yes

6. Review Comments to the Author

Reviewer #1: (No Response)

Reviewer #2: The authors have improve the manuscript to an acceptable level, the flow of information has improved significantly.

7. PLOS authors have the option to publish the peer review history of their article (what does this mean?). If published, this will include your full peer review and any attached files.

Reviewer #1: No

Reviewer #2: No

---

## [Editor Report · Acceptance letter]

7 Jun 2024

PONE-D-24-00504R1 

PLOS ONE

Dear Dr. Sundarapperuma, 

I'm pleased to inform you that your manuscript has been deemed suitable for publication in PLOS ONE. Congratulations! Your manuscript is now being handed over to our production team.

Kind regards, 

on behalf of

Dr. Trhas Tadesse Berhe 

Academic Editor

PLOS ONE